# Model-Based Fault Analysis and Diagnosis of PEM Fuel Cell Control System

Byungwoo Kang [1], Wonbin Na [2] and Hyeongcheol Lee [3,*]

1 Department of Automotive Electronics and Control Engineering, Hanyang University, Seoul 04763, Republic of Korea
2 Department of Electrical Engineering, Hanyang University, Seoul 04763, Republic of Korea
3 Department of Electrical and Biomedical Engineering, Hanyang University, Seoul 04763, Republic of Korea
* Correspondence: hclee@hanyang.ac.kr; Tel.: +82-2-2220-1685

**Abstract:** This paper presents a systematic fault analysis and diagnosis method of a PEM fuel cell control system using a model-based approach. With a model-based approach, it is possible to analyze the causal relationship and effect of probable faults in the system, and to diagnose them under the assumption that the model and the process are similar. With a model-based approach, it is possible to analyze the causal relationship and effect of probable faults in the system and diagnose them under the assumption that the model and the process are similar. In this work, a model-based approach was adopted for fault analysis and diagnosis, and its methods are suggested. A PEM fuel cell is mathematically modelled, analyzed, and verified for the analysis and simulations. Relationships among variables are shown using an incidence matrix and with a Dulmage–Mendelsohn decomposition of the matrix. When it is difficult to detect faults due to a deficient degree of redundancy, a bi-partite graph is used to analyze the effect of faults and to assess the possibility of fault detection through the appropriate redundant sensor placement. Thereafter, residuals are obtained based on analytical redundancies of the system, and a fault signature matrix is subsequently constructed. A fault detection and isolation (FDI) algorithm is developed based on a fault signature matrix that describes the connection between faults and residuals. The simulation results demonstrate the validity and effectiveness of the proposed FDI algorithm for diagnosing faults. With the proposed FDI algorithm, eight faults could be diagnosed by FDI algorithm with given sensors in the system.

**Keywords:** PEM fuel cell system fault analysis; PEM fuel cell system fault detecting algorithm; Dulmage–Mendelsohn decomposition; Bi-partite graph

## 1. Introduction

There has been ongoing research on hydrogen energy over the last few decades. "Hydrogen has long been heralded as an alternative to fossil fuel [1]". It is one of the fuels that can be used to power automobiles, since PEMFC (polymer electrolyte membrane fuel cells) can be mounted on fuel cell vehicles. Figure 1 shows the PEM fuel cell diagram. The movement towards net zero emission has accelerated since the 2015 Paris climate agreement. It is obvious that road vehicles are a major source of air pollution. Thus, automakers are shifting from traditional internal combustion engine vehicles to electric vehicles. However, generating electricity also emits pollutants, so net zero cannot be fully achieved. Green hydrogen can be considered as a solution for sustainability. Green hydrogen is a term used for hydrogen that is produced from renewable energy so that there is zero emission of carbon throughout the whole process of energy generation and consumption.

There have been several novel works on the fault diagnosis of fuel cell systems. Most of these studies were conducted with one of two methodologies: a data analysis approach or a model-based approach. Nowadays, the data analysis approach in the fault analysis and diagnostics of the PEM fuel cell through a machine-learning process is the most

promising method. The complexity of the PEM fuel cell model can be overcome using data analysis. Lim et al. presented a component-level fault diagnosis of PEM fuel cell thermal management systems through support vector machine models with temperature, pressure, and fan control signal data [2]. Mao et al. conducted research on identifying abnormal sensors during the PEM fuel cell operation by identifying the state of PEM fuel cells using the Kernel principal component analysis (KPCA) [3] and single value decomposition of multiple sensor measurements [4]. Lee et al. proposed a diagnostic method with empirically obtained residuals and the classification of the fault states with several machine learning methods [5]. Zhang et al. presented a method of extracting features from measurements with a back propagation neural network (BPNN), and the converted feature maps were deployed to realize the fault classification with a convolutional neural network (CNN) [6]. Liu et al. proposed a discrete hidden Markov model (DHMM) fault diagnosis strategy with a K-means clustering algorithm to filter the outlying sample points to solve the fault classification problems of fuel cell tramways [7]. Yang et al. developed an improved data-driven fault diagnosis algorithm for the SOFC system to solve the problems that make a fault diagnostic algorithm less implementable on the system [8]. A random forest algorithm combined with a mean impact value (MIV) index was used to extract important characteristic parameters and train the fault classifier.

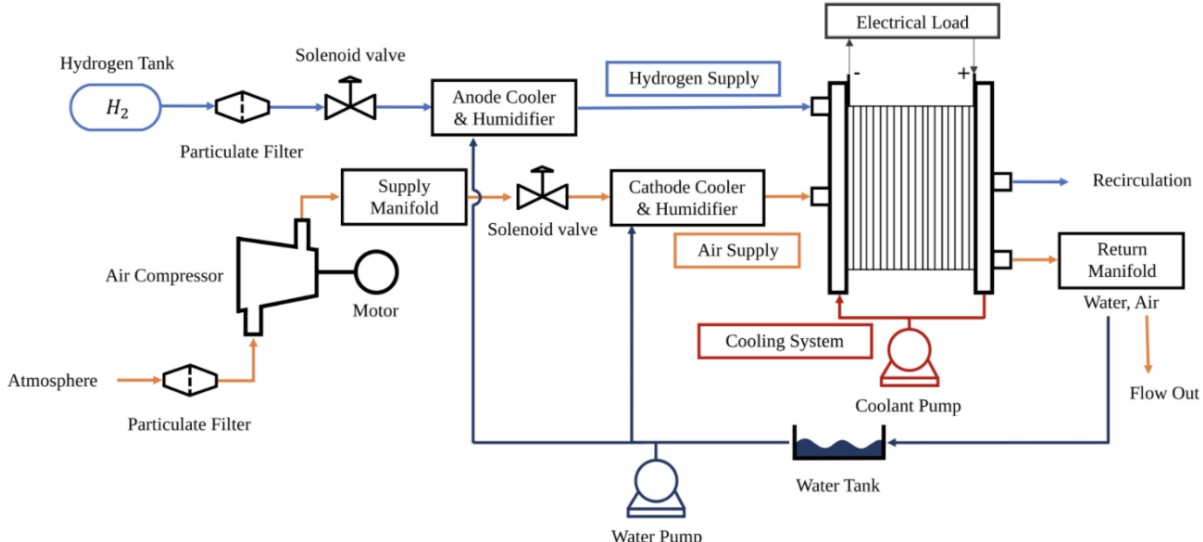

**Figure 1.** PEM fuel cell diagram.

On the other hand, a model-based analysis makes it difficult to deal with complex systems. However, when it comes to system analysis, it is more reliable since the analysis was conducted mathematically by examining the effect of fault throughout the whole system. Petrone et al. introduced studies on model-based diagnosis methodologies for PEMFCs [9]. Escobet et al. presented a model-based fault diagnosis of a PEM fuel cell by computing residuals with an analytical relationship [10]. Relative residual fault sensitivity was suggested, and the residual threshold was easily determined, regardless of the magnitude of each signal. Aitouche et al. developed an approach for deriving nonlinear analytical redundancy via parity space [11]. With this parity space approach, the authors overcame the difficulty of generating analytical redundancy of a nonlinear system compared to it being done in a linear system. Bougatef et al. and Lira et al. worked on the fault diagnosis of PEM fuel cells with a LPV (linear parameter varying) observer [12,13]. PEM fuel cell models were considered as LPV systems in both studies, and the LPV observer helped overcome difficulties in online measurement parameters, which are an online measurable state dependent parameter and a time delay dependent parameter. Rubio et al. proposed a methodology for diagnosing the performance degradation of PEM fuel cells

by making equivalent-circuit models of the PEM fuel cell impedance and proving the relationships between the parameters of the equivalent-circuit model and the electrochemical cell model [14].

In this study, we address model-based fault diagnosis and the development of fault detection and isolation for PEM fuel cell control systems. Polverino et al. and Rosich et al. conducted a model-based fault diagnosis on the PEM fuel cell system with a similar methodology, applying the Dulmage–Mendelsohn decomposition [15,16]. Polverino's research introduces the fault diagnosis of the PEM fuel cell system in a general case and proposes how fault diagnosis can be improved with different system models. In this paper, components of the PEM fuel cell control system are in consideration and assumed to have faults to be diagnosed. The work in this paper is distinguished from the work carried out by Rosich by counting in the fault of various sensors in the PEM fuel cell control system. Furthermore, this work not only applies the Dulmage–Mendelsohn decomposition technique on the system model for fault diagnosis but also investigates how variables effect the detectability of the fault diagnosis system, especially when it comes to the variables related with the air compressor map. The paper is organized as follows. First, we provide an overview of PEM fuel cell systems. Next, we discuss the fault analysis of the system and a strategy for developing FDI algorithms. Finally, FDI algorithms were implemented on simulation models and the analysis of simulation results.

## 2. PEMFC System Model and Verification

PEM fuel cells convert chemical energy into electrical energy through electrochemical reactions. Hydrogen is oxidized and split into hydrogen ions and electrons at the anode. Hydrogen ions travel across the electrolyte and electrons move through the electric circuit planted in the stack. On the cathode, they are combined with oxygen, and water is formed as a by-product. An electric potential is formed between the electrodes through the electrochemical reaction in the cell. Cells are stacked up to achieve a higher voltage and are used to construct a stack system. The PEM fuel cell system consists of a fuel cell stack and balance of plant (BoP). The BoP includes the air supply system, fuel supply system, and thermal management system. In this study, we address faults in the air supply system, fuel supply system, and fuel cell stack system. The Table 1 below shows sensors and actuators used in respective systems.

**Table 1.** List of sensors and actuators in subsystems.

| Subsystem | Type | Component |
|---|---|---|
| Air supply | Sensor | Air flow sensor |
| | | Air pressure sensor |
| | Actuator | Air compressor |
| | | Air shut-off valve |
| Fuel supply | Sensor | Fuel flow sensor |
| | | Fuel pressure sensor |
| | Actuator | Fuel flow valve |
| | | Ejector |
| Fuel cell Stack | Sensor | Voltage sensor |
| Sensor | Sensor | Coolant temp. sensor |
| | Actuator | Coolant pump |
| | | Coolant flow valve |

*2.1. PEMFC Model*

2.1.1. Air Supply System

Sufficient reactants must be provided for the fuel cell stack to produce the required amount of electric energy. Therefore, air from the atmosphere is compressed using an air compressor so that the air flows into the fuel cell stack for power generation. The air flow

rate is determined by the motor speed and the pressure difference between the atmosphere and supply manifold. Though the relationship can often be determined from a 3-D look-up table, faults cannot be mathematically analyzed with the look-up table. Therefore, in this paper, a mathematical model of the air compressor is used for the model-based fault analysis. Table 2 shows the coefficient values of air compressor equations. The equations are obtained from a surface-fitted function with Jensen and Kristensen's method from Pukrushpan's model [17].

$$M = \frac{U_c}{\sqrt{\gamma R_a T_{cp,IN}}} \tag{1}$$

$$U_c = \frac{d_c}{2\sqrt{\theta}} \omega_{cp} \tag{2}$$

$$\phi_{\max} = a_4 M^4 + a_3 M^3 + a_2 M^2 + a_1 M + a_0 \tag{3}$$

$$\beta = b_2 M^2 + b_1 M + b_0 \tag{4}$$

$$\psi_{\max} = c_5 M^5 + c_4 M^4 \atop + c_3 M^3 + c_2 M^2 + c_1 M + c_0 \tag{5}$$

$$\psi = C_p T_{cp,IN} \frac{\left[ \left( \frac{p_{cp,OUT}}{p_{cp,IN}} \right)^{\frac{\gamma-1}{\gamma}} - 1 \right]}{\left( \frac{U_c^2}{2} \right)} \tag{6}$$

$$\phi = \phi_{max} \left[ 1 - exp\left( \beta \left( \frac{\psi}{\psi_{max}} - 1 \right) \right) \right] \tag{7}$$

$$W_{cp} = \frac{\delta}{\sqrt{\theta}} \frac{\pi \phi \rho_a d_c^2 U_c}{4} \tag{8}$$

**Table 2.** Coefficient values of air compressor equations.

| Coefficient | Value |
|:---:|:---:|
| $a_4$ | $-3.69906^{-5}$ |
| $a_3$ | $2.70399^{-4}$ |
| $a_2$ | $-5.36235^{-4}$ |
| $a_1$ | $-4.63685^{-5}$ |
| $a_0$ | $2.21195^{-3}$ |
| $b_2$ | $1.76567$ |
| $b_1$ | $-1.34837$ |
| $b_0$ | $2.44419$ |
| $c_5$ | $-9.78755^{-3}$ |
| $c_4$ | $0.10581$ |
| $c_3$ | $-0.42937$ |
| $c_2$ | $0.80121$ |
| $c_1$ | $-0.68344$ |
| $c_0$ | $0.43331$ |

The air mass flow rate through a compressor $W_{cp}$ is derived from the air compressor motor speed $\omega_{cp}$, temperature of the air in the supply manifold $T_{cp,IN}$, and pressure of the air in the supply manifold $p_{cp,OUT}$, but the mass flow rate is also regulated by the air shut-off valve. $M$ is the inlet Mach number and $\phi_{\max}$, $\beta$, and $\psi_{\max}$ are polynomial functions of the $M$. $\psi$ is the dimensionless head parameter and $\phi$ is the normalized compressor flow rate. $U_c$ is compressor blade tip speed, $\gamma$ is the ratio of the specific heats of the gas at constant pressure, $R_a$ is the air gas constant, $d_c$ is the compressor diameter, $\theta$ is corrected temperature, $\delta$ is corrected pressure, and $\rho_a$ is air density. The air shut-off valve is normally open, and a fault can occur. A fault is specified as in the proportional electric valve described by Isermann [18]. The amount of air that passes through the valve is

defined as the air flow rate ratio $\phi_{air,flow}$ and is controlled by its input $u_{valve,air}$. The fault variable $f_{valve,air}$ describes the fault in the valve, and $d_1$ is the characteristic coefficient of the valve.

$$\phi_{air,flow} = (1 - f_{valve,air}) \times \left[1 + \left((1 - d_1) \times (u_{valve,air} - 1) + d_1 \times (u_{valve,air} - 1)^2\right)\right] \quad (9)$$

$$W_{air} = W_{cp} \times \phi_{air,flow} \quad (10)$$

### 2.1.2. Fuel Supply System

In the fuel supply system, the amount of fuel ejected is determined by a control algorithm. Before the fuel goes into the anode for the chemical reaction, the mass flow rate of the fuel is also regulated by the fuel flow valve. The fuel flow valve is a normally closed electrical proportional valve that controls the mass flow rate of fuel. Not enough fuel flows to the anode when the coil is burned. Therefore, the valve performance is defined as noted below, which is similar to the valve of an air supply system.

$$\phi_{H_2,flow} = (1 - f_{valve,H_2}) \times \left[1 + \left((1 - d_2) \times (u_{valve,H_2} - 1) + d_1 \times (u_{valve,H_2} - 1)^2\right)\right] \quad (11)$$

$$W_{H_2} = W_{an,IN} \times \phi_{H_2,flow} \quad (12)$$

Modeling of the hydrogen inlet pressure is determined and modeled based on a study by Zuhaili et al., which focuses on hydrogen inlet pressure parameter analysis [19].

### 2.1.3. Humidifier

Humidifiers in fuel cell systems provide water vapor to prevent too much water (which is produced from the electrochemical reaction) from being absorbed into the dry air. It prevents the cathode from drying out and the anode from suffering from low humidity because of electroosmotic drag. Meanwhile, the electrolyte membrane of the fuel cell system must also be sufficiently hydrated to maintain hydrogen ion conductivity. When the moisture in the electrolyte membrane is insufficient, the hydrogen ion conductivity decreases as it dries and causes the membrane to contract. The contraction of the membrane leads to the separation of the electrode and the membrane, and the gap between them will eventually increase the contact resistance. On the other hand, when an excess of water is supplied, it becomes difficult for the reactive gas to access the surface of the porous catalyst, and the stack performance is greatly reduced. Furthermore, since excessive humidification causes flooding in cells, it must be adequately controlled. The performances of fuel cells in different humidification conditions were studied by Kim et al. [20] They showed that both the anode and cathode had the best performance at relative humidity values of 100 percent compared to other conditions.

### 2.1.4. Fuel Cell Stack

Hydrogen and air are supplied to the anode and cathode, respectively, and they react with the electrolyte to form ions. In the process of forming water from electrochemical reactions between the generated ions, electrons are generated at the anode and move to the cathode, eventually generating electricity. An electric potential is formed between the electrodes, and the electrode potential, *E*, is determined by the Nernst equation, which is shown below.

$$E = E^0 + \frac{RT_{fc}}{2F} ln \frac{p_{H_2} p_{O_2}^{1/2}}{p_{H_2O}} \quad (13)$$

A standard reversible cell potential $E^0$ of 1.229 V was obtained. When the entropy change of the reaction is assumed to be constant, the equation can be written as:

$$E = 1.229 - 0.85 \times 10^{-3} \left( T_{fc} - 298.15 \right) + 4.3085 \times 10^{-5} T_{fc} \left[ ln\left(p_{H_2}\right) + \frac{1}{2} ln\left(p_{O_2}\right) \right] \quad (14)$$

When the cell is in operation, it is followed by three losses: activation loss, ohmic loss, and concentration loss. Therefore, the actual voltage of the fuel cell is written as follows.

$$v_{fc} = E - v_{act} - v_{ohm} - v_{conc} \quad (15)$$

### 2.2. Model Verification

It is aimed to utilize the model to analyze faults and develop a fault diagnostic algorithm. The algorithm is then implemented on PEM fuel cell simulation models to validate the fault classifying performance. A PEM fuel cell simulation model is acquired from Mathworks [21]. In fault diagnosis, the modeling error between the model for fault analysis and the model for simulation makes it difficult to diagnose faults. Therefore, the model for fault analysis is verified to see if it is appropriate for making the algorithm. The simulation model from Mathworks was considered accurate because its composition of the system was fully set. It includes a fuel processing unit, air processing unit, thermal management system, and fuel cell stack to be simulated synchronously. As shown in Figures 2 and 3, the normalized stack current and cell voltage of the simulation model and fault analysis model are shown in the graphs. Since the parameters and control logics differ from each other, normalized outputs are compared with the same electrical load on both models. The model for fault analysis shows similar trends compared to the model for simulations, and fault analysis was conducted.

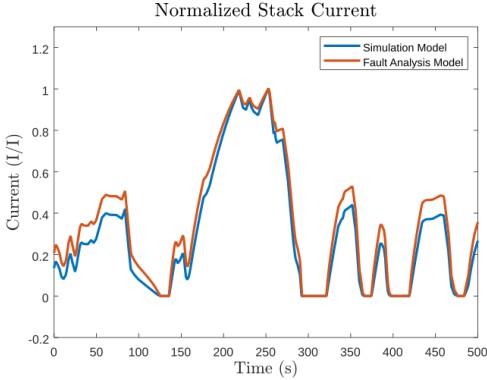

**Figure 2.** Comparison of normalized stack current between simulation model and fault analysis model.

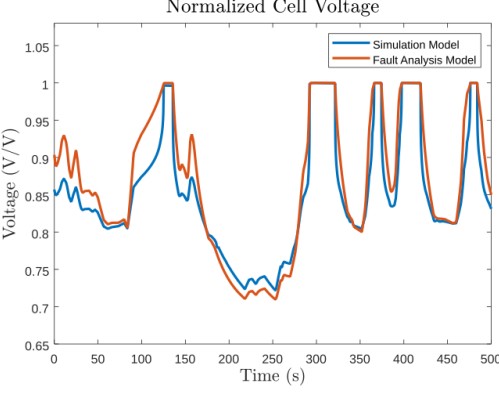

**Figure 3.** Comparison of normalized cell voltage between simulation model and fault analysis model.

### 3. Fault Analysis

Fault analysis can be conducted with a mathematical model of PEM fuel cells. When a fault occurs, it propagates through variables that are related by mathematical equations. On the basis of this principle, the direction of the fault must be clarified by examining the relationship among equations. When the values of the known variables (e.g., control input, sensor measurement) are determined, the values of unknown variables can also be obtained with equations. Therefore, an incidence matrix, which is a logical matrix used to present the relationship between variables, is used. There are known variables and unknown variables, which are the variables that cannot be measured, and faults.

After clarifying the relationship between the variables of the model, the Dulmage–Mendelsohn decomposition is applied to the incidence matrix to determine the equation set with redundancy. With the Dulmage–Mendelsohn decomposition, equations are classified into three parts: the overdetermined part, the determined part, and the underdetermined part [22]. In the overdetermined case, the number of equations is greater than the number of unknown variables by more than one. To specify the unknown variables, the same number of unknown variables and unique equations are required. When there are more equations than unknown variables, the redundant equations can be used to determine the fault. The value of the fault variable will be zero in the case where there is no fault. The number of redundant equations is called the degree of redundancy. As the degree of redundancy increases, there are more methods to obtain the values of fault variables. In the determined case, the number of equations is the same as the number of unknown variables. Therefore, unknown variables can be specified, but there are no redundant equations and there is no possibility of determining the value of the fault variable. In the underdetermined case, the number of equations is less than the number of unknown variables that are related. There is no possibility of unknown variables nor of fault variables being determined.

As the number of sensors in the system increases, the degree of redundancy tends to get higher since only a single fault is considered. When multi-fault is considered, the degree of redundancy will remain the same, even though additional sensors are applied in the system because of the possibility of the fault of added sensors. Hardware redundancy and the structural conversion of the system for increasing the number of relations of variables are the solutions to increase the degree of redundancy in multi-fault cases.

Prior to the fault analysis, components that are vulnerable to faults are chosen: faults in sensors, faults in actuators, and faults chosen from the study of the fault tree analysis (FTA) on PEM fuel cells [23,24]. Faults in sensors and actuators from Table 1 are both considered. From FTA, hydrogen leakage of PEMFCs and membrane degradation shows the highest mean failure rate among others. The failure rate was calculated from a Weibull distribution, which is used for lifespan data analysis. Furthermore, the stack voltage drop arises from PEMFC degradation. Fault equations are added into the system equation set, and fault analysis is conducted.

The incidence matrix shows the relationship between variables. With the incidence matrix, the overdetermined part of the result of the Dulmage–Mendelsohn decomposition is shown in Figure 4. Analytical redundancy exists in the overdetermined part, and single faults in this area are detectable. Faults not included in the overdetermined part were undetectable due to a lack of redundancy, and appropriate sensor placement is required to diagnose those faults.

In the overdetermined case, residuals are obtained from a bi-partite graph. Bi-partite graphs link variables and show how they are affected by other related variables. The residual is calculated from the difference between measured outputs, between estimated values of a variable, or between the measured output and estimated values. Figure 5 shows the relationship among variables in residual 5, which is derived from the overdetermined part in Figure 3. Using the same procedure, every useful residual to be adopted in forming a diagnostic algorithm is calculated and is shown in Table 3. Residual 5 and residual 6 in Table 3 are the differences between the estimated values of the normalized compressor flow rate $\hat{\phi}$ determined in different ways, as shown in Figure 6.

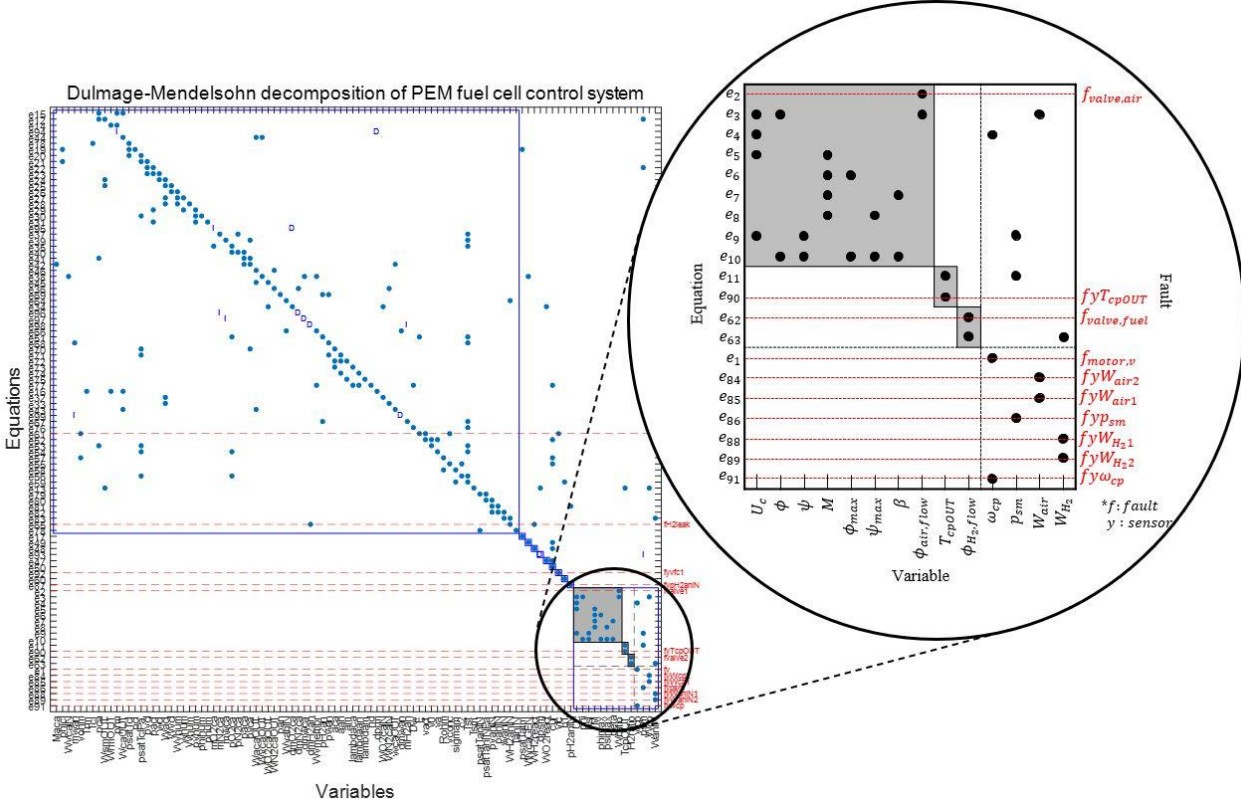

**Figure 4.** Overdetermined part of incidence matrix.

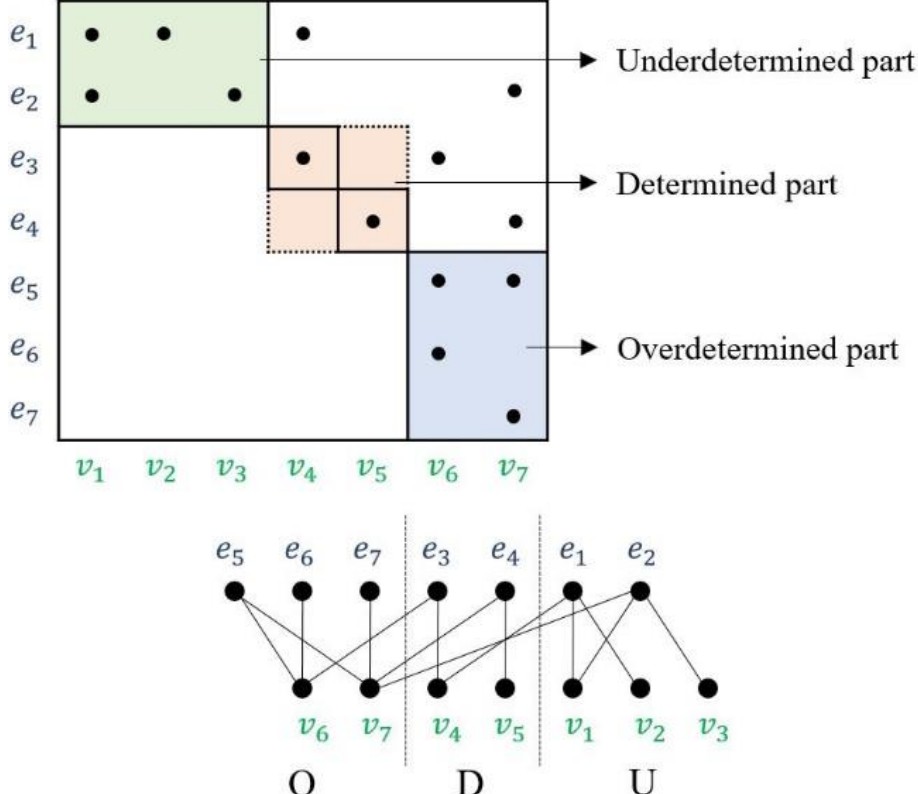

**Figure 5.** Dulmage–Mendelsohn decomposition.

**Table 3.** List of residuals obtained and respective inputs.

| Residual | Input |
|---|---|
| $R1 = W_{an,IN1} - W_{an,IN2}$ | Fuel flow sensor 1<br>Fuel flow sensor 2 |
| $R2 = W_{an,IN1} - \hat{W}_{an,IN}$ | Fuel flow sensor 1<br>Fuel valve input |
| $R3 = W_{an,IN2} - \hat{W}_{an,IN}$ | Fuel flow sensor 2<br>Fuel valve input |
| $R4 = W_{cp1} - W_{cp2}$ | Air flow sensor 1<br>Air flow sensor 2 |
| $R5 = \phi - \hat{\phi}$ | Air flow sensor 1<br>Air valve input<br>Motor speed sensor<br>Air pressure sensor |
| $R6 = \phi - \hat{\phi}$ | Air flow sensor 2<br>Air valve input<br>Motor speed sensor<br>Air pressure sensor |
| $R7 = T_{cp,OUT} - \hat{T}_{cp,OUT}$ | Air pressure sensor<br>Air temperature sensor |
| $R8 = \omega_{cp} - \hat{\omega}_{cp}$ | Motor speed sensor<br>Motor input |
| $R9 = W_{cp} - \hat{W}_{cp}$ | Air flow sensor 1<br>Air valve input<br>Air pressure sensor<br>Motor input |
| $R10 = W_{cp} - \hat{W}_{cp}$ | Air flow sensor 1<br>Air valve input<br>Motor speed<br>Air temperature sensor |

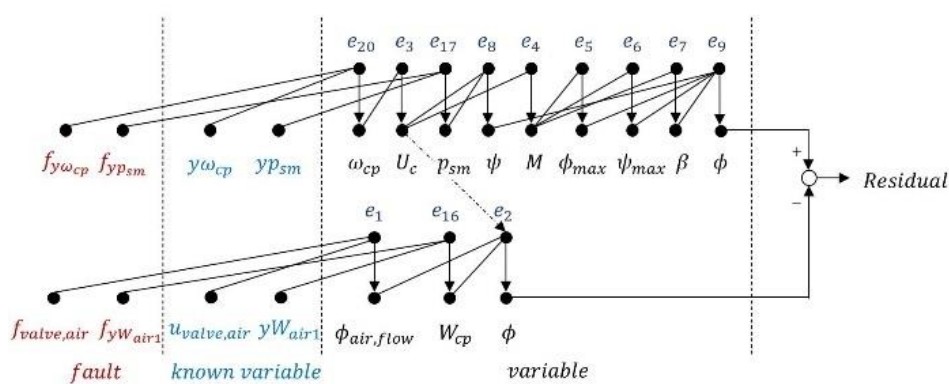

**Figure 6.** Bi-partite graph of equation set with analytical redundancy.

When one actuator fault is considered in the air supply system, it is clear whether the fault is from the sensor or actuator due to the redundancy of the system. However, when two actuators (air compressor motor and air flow valve) are considered, the system loses its redundancy, and the faults are not able to be isolated. The reason for this is that it is impossible to define whether the mass flow rate changed as a result of a valve fault or because the measured value deviated from the real value. To solve this problem, an additional flow sensor is needed. Valve faults and sensor faults can be isolated by adding a redundant flow sensor in the system as a hardware redundancy to detect sensor fault. Additionally, a redundant fuel flow sensor is added in the fuel supply system due to the

low degree of redundancy of the system. Thus, there are two air flow sensors and two fuel flow sensors to diagnose the fault between sensors and actuators with given sensors in the system.

## 4. FDI Algorithm

In the obtained residuals, a unique combination of faults is developed to obtain the fault signature matrix, as constructed in Figure 7. In the fault signature matrix, the combination of residual signals makes it possible to classify the fault, and a fault detection and isolation algorithm is developed.

| Residual \ Fault | $fyW_{H_2}1$ | $fyW_{H_2}2$ | $f_{valve,fuel}$ | $fyW_{air1}$ | $fyW_{air2}$ | $f_{valve,air}$ | $fy\omega_{cp}$ | $fyp_{sm}$ | $f_{motor,v}$ | $fyT_{cp,OUT}$ |
|---|---|---|---|---|---|---|---|---|---|---|
| Residual 1 | 1 | 1 | | | | | | | | |
| Residual 2 | 1 | | 1 | | | | | | | |
| Residual 3 | | 1 | 1 | | | | | | | |
| Residual 4 | | | | 1 | 1 | | | | | |
| Residual 5 | | | | 1 | | 1 | 1 | 1 | | |
| Residual 6 | | | | | 1 | 1 | 1 | 1 | | |
| Residual 7 | | | | | | | | 1 | | 1 |
| Residual 8 | | | | | | | 1 | | 1 | |
| Residual 9 | | | | 1 | | 1 | | 1 | 1 | |
| Residual 10 | | | | 1 | | 1 | | | 1 | 1 |

**Figure 7.** Fault signature matrix.

According to the result of the fault analysis, the nonlinearity of the air compressor map makes it difficult to detect faults. In the map shown in Figure 8, the motor speed and pressure ratio of the supply manifold inlet air pressure and outlet pressure are inputs, and the air flow rate is the output. When there was a fault in the pressure variable, the fault must be propagated to the air flow rate variable and to the residual. However, the air flow rate remains the same regardless of the pressure ratio when the motor speed is over 70,000 rpm, and this problem raises the possibility of a fault not being detected by the residual. In this case, an adequate FDI algorithm must be implemented to overcome the problem. Furthermore, the sensitivity of each fault must be considered, and it is expressed as the ratio between the amount of change in the residual and the amount of change in the fault.

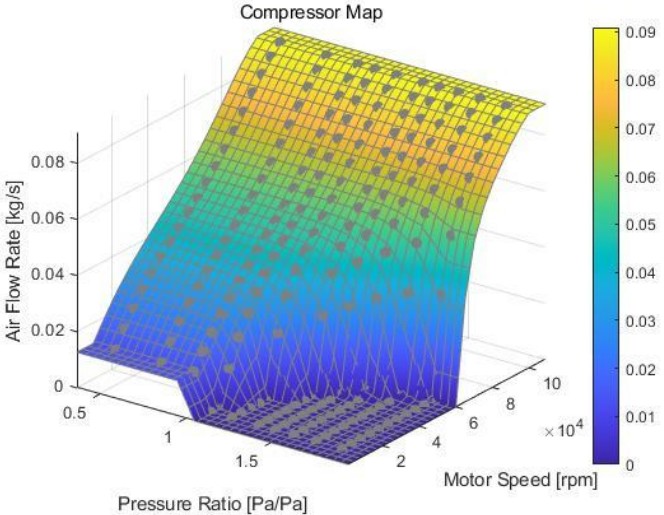

**Figure 8.** Compressor map.

When the sensitivity of the residual is high, the fault tends to be detected more easily than when the sensitivity is low. For a better classification of the faults, a FDI algorithm is constructed with a top-down method based on the sensitivity of residuals. The flow of the FDI algorithm is presented in Figure 9. The threshold of each residual is chosen empirically rather than by following a general guideline since sensor noise tended to be amplified while being delivered over variables and showing residual overlap.

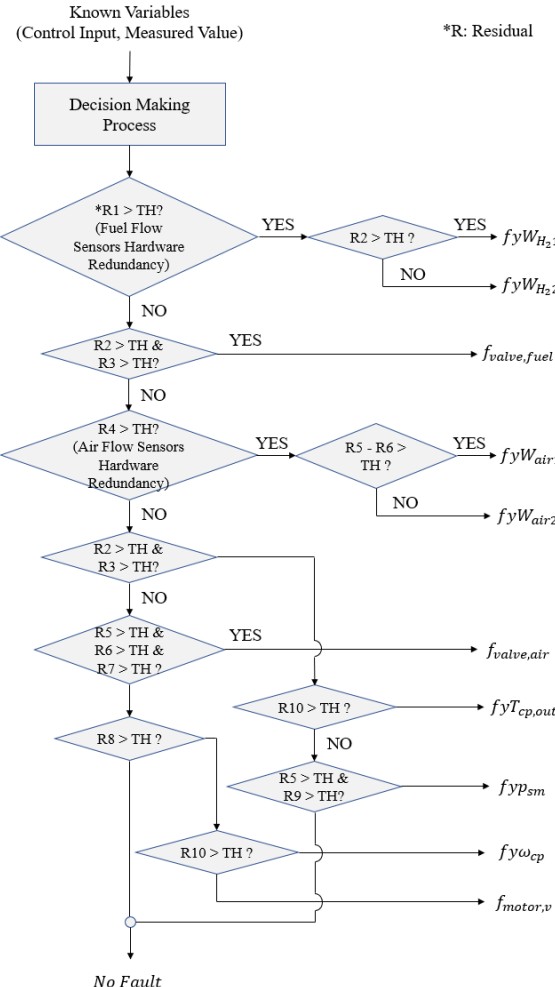

**Figure 9.** Fault detection and isolation algorithm.

## 5. Simulation Results

To validate the performance of the FDI algorithm, the simulation environment was arranged with a PEM fuel cell model for simulation and FDI algorithm implementation. Faults in sensors and actuators were defined for the simulation.

Faults in sensors were defined as additive faults resulting in an abnormal dc offset. In actuators, faults were defined as follows: additive faults in the fuel flow valve caused by weakening in the spring tension, additive faults in the air shut-off valve caused by overheating, burn out coils and multiplicative faults of minus 10 percent caused by demagnetization in overheated permanent magnets. The result of the air pressure sensor fault is shown in Figure 10. At t = 70 s, the air pressure sensor fault was set to $-3000$ Pa ($fyp_{sm} = -3000$). Through the FDI algorithm in the decision-making process, the fault was classified as an air pressure sensor fault and is shown on fault indicators in Figure 11. The proposed FDI algorithm shows the capability of diagnosing faults in the case of pressure sensor faults and in every other fault case.

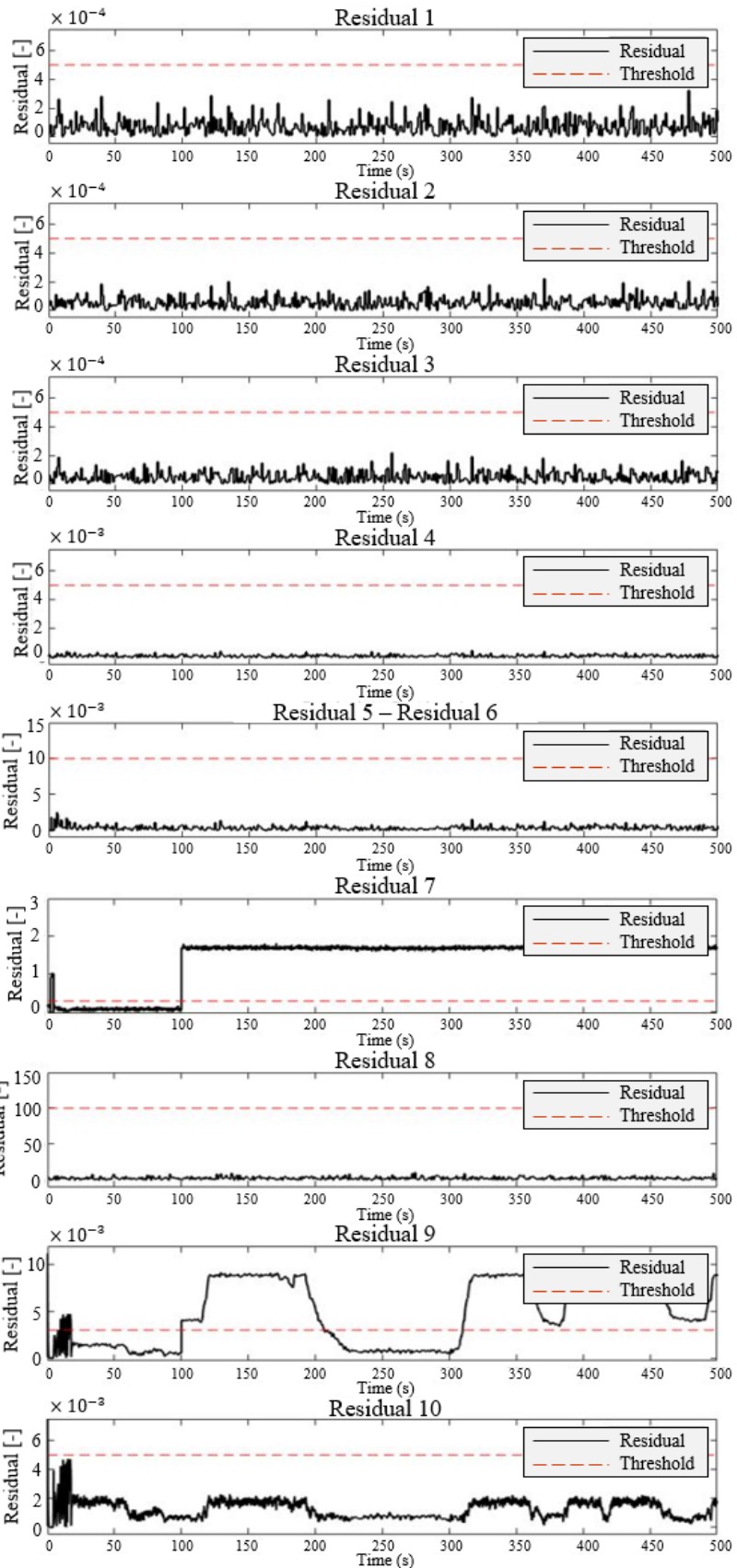

**Figure 10.** Residuals in case of air pressure sensor fault.

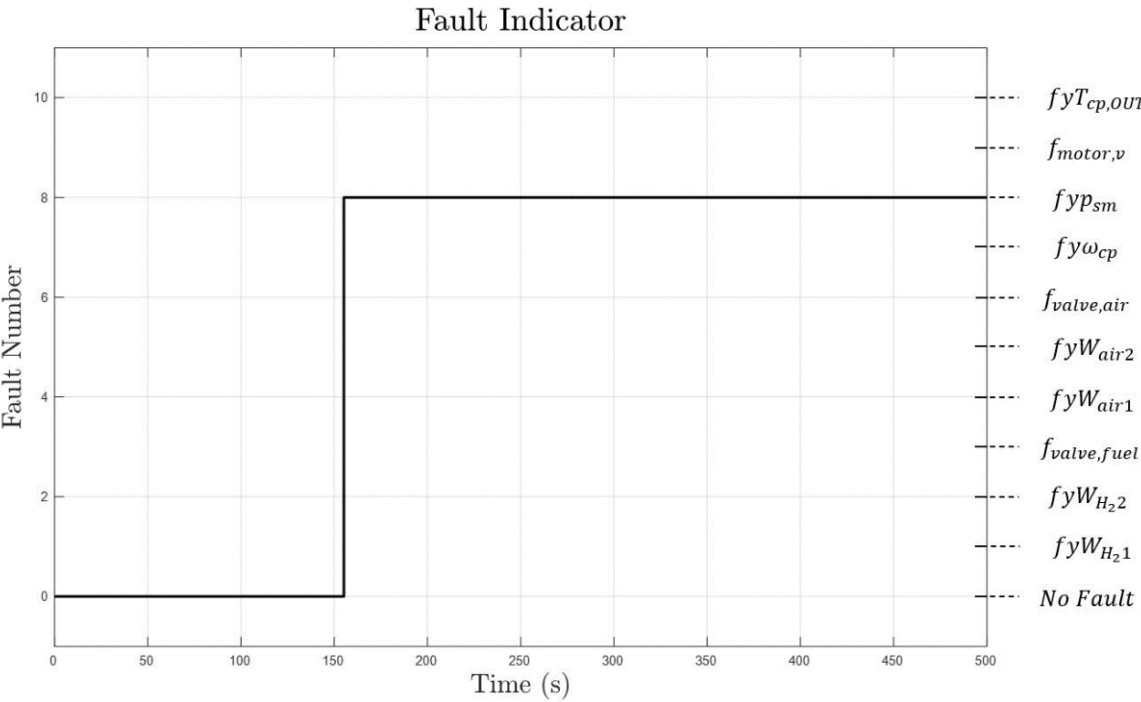

**Figure 11.** Fault indicator in case of air pressure sensor fault.

As mentioned in the fault analysis section, residual 9 in Figure 10 goes below the threshold value in the presence of a fault because of the compressor map characteristic. Therefore, the fault diagnostic algorithm must be designed to not misjudge the state of the system.

## 6. Conclusions

In this paper, a study of the model-based fault analysis and diagnosis of PEM fuel cells was conducted. The PEM fuel cell system model, including sensors and actuators, was defined and faults in the system were analyzed. The analysis of possible faults in the system could be carried out with the idea of sorting out the related variables in the system. Dulmage–Mendelsohn decomposition was a powerful method of reordering variables with relativeness. Variables in overdetermined parts were candidates of faults that could be detected. Based on the information from the fault analysis, residuals from known variables were calculated via a bi-partite graph, and a FDI algorithm was developed. The algorithm could be made much simpler with the fault signature matrix by separating variables into relative sets. The algorithm was validated on a PEM fuel cell model for simulation by being implemented in a simulation environment. As a result, residuals could detect changes in the system and the FDI algorithm could determine the fault. Eight faults (except for hardware redundancy) could be diagnosed by the FDI algorithms with given sensors in the system.

The model-based method of designing a FDI algorithm throughout the paper had proven its effectiveness by sorting out faults in the PEM fuel cell system. Although the machine-learning method can overcome the complexity of the system in diagnosing fault, it still has its limit in the analysis of the target system. This particular method had its significance especially in closely examining the PEM fuel cell system since it has a complicated relation of variables, such as flow rate, pressure, temperature, and many others. Furthermore, the algorithm can be easily modified in case there is a change of components in the system based on the information about the system studied. The methodology shown in the paper has its power in having a deeper look into how components of the system affect one another more than other methods of detecting fault.

**Author Contributions:** Formal analysis, B.K. and W.N.; project administration, B.K., W.N. and H.L.; resources, B.K. and W.N.; software, B.K. and W.N.; validation, B.K. and W.N.; writing—original draft preparation, B.K.; writing—review and editing, B.K. and H.L.; supervision, H.L. All authors have read and agreed to the published version of the manuscript.

**Funding:** This research received no external funding.

**Institutional Review Board Statement:** Not applicable.

**Informed Consent Statement:** Not applicable.

**Data Availability Statement:** Not applicable.

**Acknowledgments:** This work was supported by the Korea Institute of Energy Technology Evaluation and Planning (KETEP) grant funded by the Korea government (MOTIE) (20223030030010, Development and demonstration of optimized fuel-cell hybrid system technology for hydrogen bus). This paper is the result of research carried out by a research fund and technical support from HMC.

**Conflicts of Interest:** The authors declare no conflict of interest.

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
