# Peer review of "Model-Based Fault Analysis and Diagnosis of PEM Fuel Cell Control System"

_applsci, doi:10.3390/app122412733_

Round 1
Reviewer 1 Report
The work is related to the diagnostic of the PEMFC operation by using a model-based approach. The model used in this work is based on those developed in [15] , and is widely accepted in the control community as a good representation of the behavior of an FCS system.
This model was analyzed by means of Structural Analysis theory, applying Dulmage-Mendelsohn decomposition. Residuals are obtained based on analytical redundancies of the system and a fault signature matrix is generated.
Even this technique is interesting, and seems to be an important tool in model-based diagnostic of the FCS, such as the method described in
· Journal of Power Sources 357 (2017) 26-40.
· Applied Mathematical Modelling 38 (2014) 2744–2757
1.The presented model is not presented rigorously: many of the parameters implied in equations (1)-(8) are not explained in the text. Authors can include also the entire model in an annex.
2. In 2.2 paragraph, to verify the accuracy of the model authors consider a Mathworks simulation model as a reference. Authors must justify why this simulation model can be considered as “very accurate” to be used in the validation.
3. Flowchart in figure9 is not visible
4.Conclusions paragraph does not resume the methodology and the results.
Reviewer 2 Report
This paper investigates a novel fault diagnosis method of a PEM fuel cell 10 control system bases on a model-based approach. There are still some problems need to be addressed.
1) There are many judgment thresholds in the verification of the proposed method. How are the numerical values of these thresholds determined?
2) According to the content of the paper, the superiority of the proposed method cannot be fully understood. After all, there are already many other methods in the literature, and further explanations should be made on the advanced types and characteristics of the proposed methods. 3) The resolution of the illustrations in the paper is not high, and the illustrations with better quality should be improved.
4) The three keywords of the paper are not related to the method studied, which is not appropriate.
5) In Equations 1 to 7, it seems that the meanings of a large number of variables are not explained.
Reviewer 3 Report
Reviewer #: Model-Based Fault Analysis and Diagnosis of PEM Fuel Cell Control System. This work has a clear modernity and originality, compared to the works carried out in this field.
The article is a well written and seems to be free of technical errors. However, the paper needs a revision, as there are a few things to be checked and/or corrected. They are given below:
1. The abstract should be more clearly presents objects, methods and results; I suggest you do add some other numerical values in the conclusions and abstract in order to quantify the results.
2. There are some different methods of fault analysis and diagnostics PEM fuel cell and the most promising method is machine-learning control. The authors are advised to further review the literature by discussing some of the relevant studies and highlighting the contributions of the current study.
3. The authors are suggested to compare the performance of the proposed technique with different method in the literature.
4. The authors should further clarify the proposed FDI method with the need to highlight its efficiency through clear criteria that demonstrate the validity and effectiveness of the fuel cell fault diagnosis algorithm.
5. Figures 9 and 10' are not clear; please re-insert this figures clearly.
6. I think that the conclusion should be reformulated in a precise manner and supported by the content of the conclusions reached with reference to the efficiency of the proposed approach in order to quantify the obtained results.

Round 2
Reviewer 1 Report
Article still need some bibliographic references on FC model based diagnostic aplying Dulmange Mendhelson decomposition.
Reviewer 2 Report
It can be accepted.
Author Response
Thank you for your valuable comments and suggestions. I modified this paper based on your comments. I really appreciate your efforts and I am sure your suggestions made this paper more valuable.
Reviewer 3 Report
I see that the authors have fulfilled all the recommendations and therefore this paper can be accepted for publication in Applied Sciences journal.

Author Response

(The authors gave the same response as above.)
